# Modeling Effect of Bubbles on Time-Dependent Radiation Transfer of Microalgae in a Photobioreactor for Carbon Dioxide Fixation

**Tianhao Fei [1,2], Li Lin [1,2], Xingcan Li [3], Jia-Yue Yang [1,2,*] , Junming Zhao [4] and Linhua Liu [1,2,*]**

1  School of Energy and Power Engineering, Shandong University, Jinan 250061, China
2  Optics & Thermal Radiation Research Center, Institute of Frontier and Interdisciplinary, Shandong University, Qingdao 266237, China
3  College of Energy and Power Engineering, Northeast Electric Power University, Jilin City 132012, China
4  School of Energy Science and Engineering, Harbin Institute of Technology, Harbin 150001, China
*  Correspondence: jy_yang@sdu.edu.cn (J.-Y.Y.); liulinhua@sdu.edu.cn (L.L.)

**Abstract:** Microalgae are considered one of the most efficient and environmentally friendly ways for carbon dioxide fixation. The bubbles play an important role in analyzing the radiation transfer in photobioreactors during microalgae growth. Herein, *Chlorella* sp. and *Scenedesmus obliquus* were cultured in the airlift flat plate photobioreactor and evaluated for the temporal evolution of radiation characteristics. A one-dimensional model of bubbles on time-dependent radiation transfer in a photobioreactor was proposed, and it was well verified with the experimental result. The results indicated that with the increase of bubble volume fraction or the decrease of bubble radius, the local irradiance increased at the illuminated surface of the microalgal culture and was attenuated more rapidly along with the radiation transfer. The average specific growth rate of microalgae decreases as bubble volume fraction increases or bubble radius decreases. The volume fraction of 0.003 and a radius of 3.5 mm are the optimal operating conditions in this study for microalgae growth and carbon dioxide fixation. The presented analysis would facilitate the design and optimization of the optical and aeration configurations of photobioreactors for carbon dioxide fixation.

**Keywords:** carbon dioxide fixation; microalgae; time-dependent radiation characteristics; $CO_2$ bubbles; photobioreactor; light transfer

## 1. Introduction

Biological photosynthesis is considered one of the most viable ways for carbon dioxide capture and storage [1]. Compared with terrestrial plants, microalgae exhibit the advantages of high photosynthetic efficiency, strong environmental adaptability and no major competition for agricultural land [2,3]. Biomass components, such as carbohydrates, proteins and fatty acids [4], and its valuable metabolites, such as pigments and antibiotics, offer a wide range of applications and arouse extensive attention [5]. Nowadays, large-scale cultivation of microalgae is mainly carried out in open ponds that require large spaces. However, closed-system photobioreactors (PBRs) allow the maintenance of optimal conditions for growth, making them more efficient than open systems [6]. The utilization of light energy by microalgae is a crucial factor for biomass productivity and cultivation efficiency [7,8]. The radiation characteristics of microalgae vary during the growing period, which will affect the light utilization efficiency. Sparging with bubbles not only enables effective mixing within the PBR but also improves gas-liquid mass transfer [9–11]. However, the presence of bubbles affects the light transfer within the PBRs. Therefore, a comprehensive analysis of the radiation transfer in PBRs during the growth of microalgae considering the effect of bubbles is instructive to design and optimize the light distribution and aeration to achieve a high growth rate and promote carbon dioxide fixation.

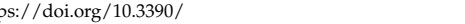

The Lambert–Beer law and two-flux method were widely used to simulate the local fluence rate in PBRs due to their simplicity [12–15]. Note that it can lead to large errors in predicting the fluence distribution where multiple scattering dominates. Pruvost et al. [16] theoretically calculated the radiative transfer in a solar rectangular PBR, considering dynamic solar radiation combined with growth kinetics. Huang et al. [17] studied the light distribution in the internally irradiated cylindrical PBR. The reflection of walls was found to play an important role in light distribution. However, an empirical reflectance was adopted in the model, and the scattering was approximated as isotropic. Lee et al. [18] investigated the light distribution coupled with growth kinetics in open ponds and PBRs exposed to time-dependent solar irradiance. Kandilian et al. [19] investigated the radiation characteristics of microalgae influenced by the growth conditions using both theoretical and experimental methods. The coated sphere approximation agreed well with the measured integral radiation characteristics of *Chlorella vulgaris*. It is noted that these studies did not take into account bubbles and time-dependent radiation characteristics of microalgae.

Some works have considered the time-dependent radiation characteristics to simulate light distribution in PBRs. Some researchers [20,21] uncovered diel variations in the scattering and absorption cross-sections of microalgae. Pilon's group [22] reported the time evolution radiation characteristics of *Nannochloropsis oculata* over full growth phases, which were found to vary significantly in response to changes in fluence rate and nutrient availability. Moreover, Pilon's group [23] investigated the mass absorption and scattering cross-sections of *Anabaena cylindrica* on 5 different days and found that the trends of different absorption peaks were quite different from each other. Zhao et al. [24] measured the time-dependent radiation characteristics of the three species of microalgae and showed that the absorption and scattering cross-sections generally decreased with cultivation time. Ma et al. [25] determined the time-dependent radiative properties of *C. vulgaris* based on the Lorenz–Mie theory in combination with the growth-dependent cell size distributions and pigment content and indicated that large errors will be introduced when cell growth is neglected. In previous studies, the influence of $CO_2$ bubbles on the radiation characteristics of microalgae during the growing period has been neglected.

A clear growth enhancement is generally observed when supplying an air stream containing $CO_2$ [26–28]. Furthermore, some studies [29–31] reported that reducing the size of $CO_2$ bubbles can significantly improve the growth of microalgae in PBRs. However, that research focused on the gas-liquid mass transfer rather than the changes and effects of radiative transfer with aeration. Pilon's group [32] first proposed a model accounting for anisotropic scattering by both the bubbles and microorganisms and considered the spectral radiation characteristics. It was concluded that the Lambert–Beer law cannot be applied to predict the irradiance and that anisotropic scattering by the bubbles should be considered. They also obtained the radiation characteristics of several species of microalgae from normal-normal and normal-hemispherical transmittance measurements and a polar nephelometer [33,34]. Wheaton and Krishnamoorthy [35] simulated the 3D distributions of radiation coupled with fluid hydrodynamics in the PBRs and found that 1-μm-size bubbles more effectively redistribute the radiation downstream of the radiators. McHardy et al. [36] numerically investigated the impact of gas bubbles on light distribution in a bubble column PBR under different gas flow rates and microalgae concentrations. Luzi et al. [37] evaluated the enhancement of culture growth by pulsed illumination and pneumatic mixing in a bubble column PBR through numerical simulations. The above studies revealed that it was necessary to consider bubbles in the radiation transfer calculation for PBRs. However, the effects of bubble parameters, including bubble volume fraction and size, have not been well studied.

In this work, two typical species of $CO_2$ fixation microalgae, *Chlorella* sp. and *S. obliquus*, were selected for experimental cultivation in an airlift flat plate PBR. An improved transmission method was conducted to determine the time-dependent radiation characteristics of microalgae. An accurate radiation transfer model considering microalgal time-dependent radiation characteristics, bubble scattering and wall surface reflection was verified to pre-

dict radiation distribution in the PBR during the growing period. The local irradiance in the PBR for time-dependent radiation characteristics was obtained and compared with the stationary results. An investigation of bubble volume fraction and radius was also carried out.

## 2. Experiments and Methodology

### 2.1. Cultivation and Sample Preparation

To cultivate microalgae, an airlift flat panel of polymethyl methacrylate (PMMA) PBR (produced by Shanghai Guangyu Biological Technology Co., Ltd., Wenzhou, China) with an inner length of 58 cm, a width of 8 cm and a height of 58 cm was chosen. As presented in Figure 1, 15 columns of LED lamps were parallelly installed on one side, and a photometric sensor was set on the other side to measure the intensity of transmitted light. The temperature of cultures, the intensity of LED lamps and the inflow velocity of air and $CO_2$ can be controlled with supporting equipment.

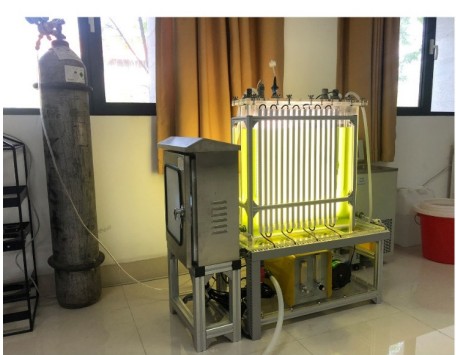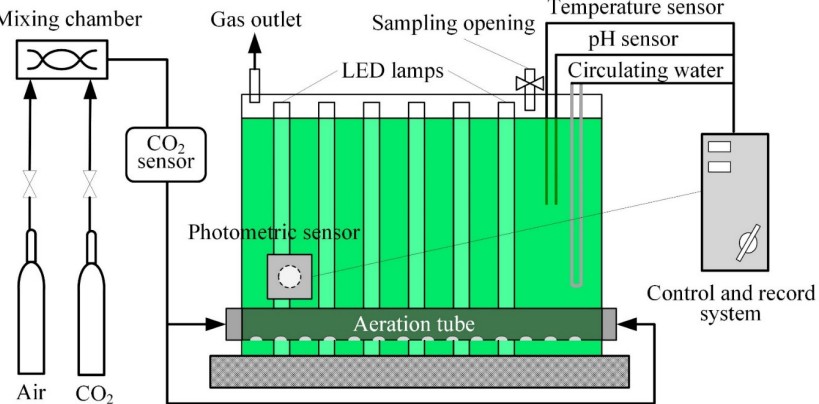

**Figure 1.** The picture and schematic diagram of the airlift flat panel PBR. (The culture temperature is controlled by heat exchange with circulating water).

*Chlorella* sp. and *S. obliquus*, two typical $CO_2$ fixation and lipid accumulation species of green algae with high $CO_2$ tolerance, were selected for cultivation [38–40]. The BG11 medium [41] was used for microalgal culture, whose composition is listed in Supplementary Materials. The medium was sterilized via heating in a high-pressure vessel, and the pH value was adjusted to 7.1 with HCl or NaOH before cultivation. The microalgae were precultured in 250-mL conical flasks and subcultured in the PBR with a total culture volume of 24.6 L. The culture temperature was kept constant at 25 °C. The sparging air was mixed with pure $CO_2$ to give a concentration of 1% at a total rate of 2 L/min. The transmitted light was about 4300 lx with only BG11 medium in the PBR after adjusting light intensity. The working cycle of illumination and aeration was 12 h on and 12 h off. A black cloth was covered outside of the PBR to avoid the influence of ambient light.

A direct microscopic count was performed on the sample of microalgal culture using a plankton-counting chamber (counting area $20 \times 20$ mm$^2$, volume 100 μL) and an optical microscope (model UB203i) equipped with a CCD camera. The size of the microalgae was obtained by analyzing the microscopic images with the software ImageView. Microalgal biomass was determined by measuring the dry weight: Filter a certain volume of culture and then weigh the microalgae after drying at 105 °C for 8 h. About 500 mL of culture was sampled per day for measurements, and the same volume of fresh and sterilized BG11 medium was added to the culture after sampling. The sampling volume was small enough and barely affected the status of culture in the PBR.

### 2.2. Radiation Characteristics Measurements

The time-dependent radiation characteristics of microalgae are fundamental for radiation transfer analysis in the PBR. However, the simplified optical model used in the

traditional approach to obtain the extinction coefficient omits high-order transmission [42]. Here, an improved transmission method was conducted to determine the extinction and absorption coefficients of microalgae suspensions [43]. A certain amount of culture was sampled into a cuvette to measure the normal-normal transmittance and the normal-hemispherical transmittance. Subsequently, the extinction coefficient $\beta$ and absorption coefficient $\alpha$ were calculated by [44]:

$$\beta = -\frac{1}{L_2} \ln\left( \frac{-t_1 t_3 + \sqrt{t_1^2 t_3^2 + 4 T_{\text{EXP}}^2 r_3 r_1'}}{2 T_{\text{EXP}} r_3 r_1'} \right), \tag{1}$$

$$\alpha = -\frac{1}{L_2} \ln\left( \frac{-t_1 t_3 + \sqrt{t_1^2 t_3^2 + 4 T_{\text{h,EXP}}^2 r_3 r_1'}}{2 T_{\text{h,EXP}} r_3 r_1'} \right), \tag{2}$$

where $T_{\text{EXP}}$ is the normal-normal transmittance and $T_{\text{h,EXP}}$ is the normal-hemispherical transmittance. $L_2$ is the thickness of layer 2 (liquid layer). $t_1$ and $t_3$ represent the transmittance of layer 1 (incident glass) and layer 3 (outgoing glass) of the cuvette from the incident side, respectively, $r_1'$ and $r_3$ represent the reflectance of layer 1 from the nonincident side and layer 3 from the incident side, respectively. $t_1$, $t_3$, $r_1'$ and $r_3$ are given as follows [42]:

$$t_1 = \frac{\tau_{01} \tau_{12} e^{-\alpha_1 L_1}}{1 - \rho_{10} \rho_{12} e^{-2\alpha_1 L_1}}, \; t_3 = \frac{\tau_{23} \tau_{30} e^{-\alpha_3 L_3}}{1 - \rho_{32} \rho_{30} e^{-2\alpha_3 L_3}}, \tag{3}$$

$$r_1' = \rho_{21} + \frac{\tau_{21} \tau_{12} \rho_{10} e^{-2\alpha_1 L_1}}{1 - \rho_{12} \rho_{10} e^{-2\alpha_1 L_1}}, \; r_3 = \rho_{23} + \frac{\tau_{23} \tau_{32} \rho_{30} e^{-2\alpha_3 L_3}}{1 - \rho_{32} \rho_{30} e^{-2\alpha_3 L_3}}, \tag{4}$$

where $\rho_{ij}$ and $\tau_{ij}$ represent the reflectivity and transmissivity at the interface between two neighboring media, i and j. $L_1$ and $L_3$ are the thicknesses of layer 1 and layer 3, respectively. Moreover, $\alpha_1$ and $\alpha_3$ are the corresponding absorption coefficients. The single scattering is assumed to prevail due to the quite small volume fraction of microalgae (generally < 0.001) in our experiments. Then, the average absorption and extinction cross-sections of microalgae were obtained by dividing the absorption and extinction coefficients by the cell number density, and the extinction cross-section minus the absorption cross-section is the scattering cross-section.

The normal-normal transmittance was measured by the spectroscopic ellipsometer (model RC2-DI; J.A. Woollam Co., Inc., Lincoln City, NE, USA) with a spectral range of 193–1690 nm. The normal-hemispherical transmittance was determined using an integrating sphere (model RTC-060-IG; Labsphere, Inc., North Sutton, NH, USA) with a measuring system (model Omni-DR830-SDU; Zolix Instruments Co., Ltd., Beijing, China) based on the lock-in amplifier and monochromator and ranged from 400 to 1100 nm. The optical constants of $H_2O$ and $SiO_2$ were obtained from refs. [45,46], respectively. The thickness of cuvette glass is 1.63 mm, and the optical path is 9.80 mm. In addition, a multi-angle polarized light scattering meter (LISST-VSF; Sequoia Scientific, Inc., Bellevue, WA, USA) covering the angular range of 0.1–150° at wavelength 515 nm was employed to measure the scattering phase functions of microalgae.

### 2.3. Radiation Transfer Modeling of the PBR

The radiation transfer model considering microalgal time-dependent radiation characteristics, bubble scattering and wall surface reflection was established to predict radiation distribution in the PBR. Microalgae and bubbles are uniformly distributed and randomly oriented in general due to the agitation caused by continuous aeration from the bottom. So, the cultures are assumed to be homogeneous, absorbing, scattering and non-conducting.

As shown in Figure 2, in a suspension, the radiation intensity $I_\lambda(r,\hat{s})$ in direction $\hat{s}$ and wavelength $\lambda$ at location $r$ can be expressed by the steady-state radiation transfer equation (RTE) and written as [47]:

$$\hat{s} \cdot \nabla I_\lambda(r,\hat{s}) = -\beta_\lambda I_\lambda(r,\hat{s}) + \kappa_\lambda I_b(r,\hat{s}) + \frac{\sigma_{s,\lambda}}{4\pi} \int_{4\pi} I_\lambda(r,\hat{s}_i) \Phi_\lambda(\hat{s}_i,\hat{s}) d\Omega_i, \tag{5}$$

where $\kappa_\lambda$ and $\sigma_{s,\lambda}$ are the effective spectral absorption and scattering coefficients, and the extinction coefficient is defined as $\beta_\lambda = \kappa_\lambda + \sigma_{s,\lambda}$. The scattering phase function $\Phi_\lambda(\hat{s}_i,\hat{s})$ represents the probability of light in direction $\hat{s}_i$ and solid angle $d\Omega_i$ scattering to the direction $\hat{s}$ and solid angle $d\Omega$.

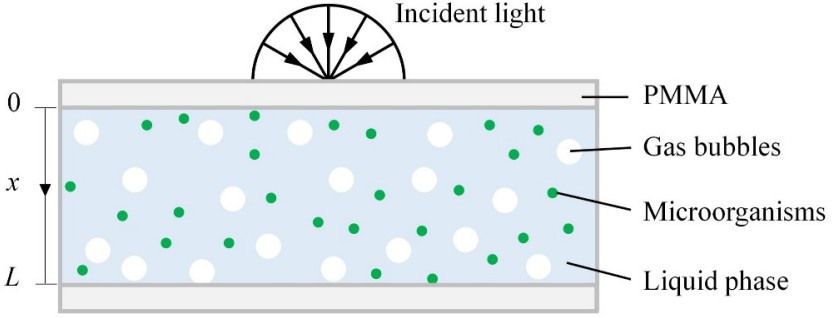

**Figure 2.** Schematic of the PBR system.

The finite volume method (FVM) was employed to discretely solve the RTE, assuming that the light transfer is one-dimensional (see computational details in Refs. [48,49] and a brief description in Supplementary Materials). The emission of the dispersive medium and walls is ignored because of the relatively low temperature. The phase functions of the bubbles and microalgae are approximated as Henyey–Greenstein (H–G) phase functions $\Phi_{HG}$ [50]. The expressions of $\kappa_\lambda$, $\sigma_{s,\lambda}$ and $\Phi_{HG}$ are shown in Supplementary Materials.

The incident light was assumed to be diffuse, and the boundary conditions were also regarded as diffuse reflection since the light transferring in the suspension illuminated the PMMA walls in different directions. The light source irradiance and boundary reflection are given in the Supplementary Materials. A box model [47] was used in which the absorption and scattering spectrum at 400–1100 nm was divided into four sections according to the position of the absorption peak. Each sub-spectrum of absorption and scattering is then approximately represented by its mean value; the light source irradiance spectrum and liquid phase absorption spectrum are also divided into four corresponding sub-spectrum in the same way; the liquid phase is cold, absorbing, and non-scattering, whose optical properties are regarded as those of pure water; the bubbles are spherical, and their scattering properties do not vary appreciably within the spectrum considered; the bubbles and microalgae are monodisperse and independent scattering prevails. In addition, the effective incident irradiance $G_{in}$, average single scattering albedo $\omega_{eff}$ and the interfacial area concentration of bubbles $A_b$ are described in Supplementary Materials.

## 3. Results and Discussion

### 3.1. Cell Growth

Figure 3 presents the growth curves of *Chlorella* sp. and *S. obliquus* plotted in cell number density and biomass mass concentrations. The curves show three typical growth phases, namely the lag phase, the exponential phase and the stationary phase. The lag phase is characterized by slow initial growth of microalgae, which was attributed to the adaptation to the new growth environment of microalgae incubated from the conical flask to the PBR. The exponential phase is characterized by a large growth rate that corresponds to the cultivation time from day 6 to day 13 for *Chlorella* sp. and from day 6 to day 15 for *S. obliquus*. The stationary phase, where the growth rate slowed down, finally occurred due to the insufficiencies of lights, nutrients or $CO_2$. The fluctuations of the microalgal growth

curve result from the wall-adhering growth of microalgae, the deposition of low-activity or dead cells, and the experimental uncertainty. During sampling, the cells depositing on or adhering to the PBR walls are not collected to reduce the influence on measurements of microalgal radiation characteristics.

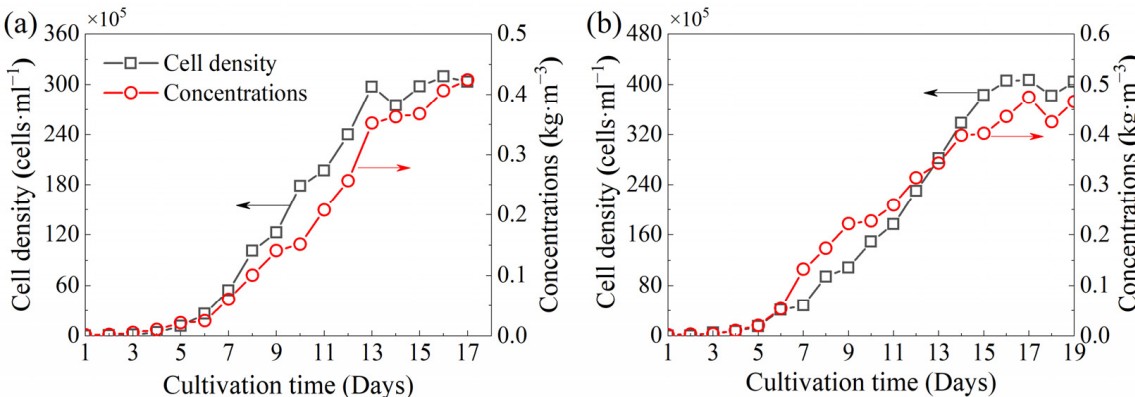

**Figure 3.** Cell number density and mass concentrations of (**a**) *Chlorella* sp. and (**b**) *S. obliquus* with different cultivation times.

Figure 4 shows micrographs and cell size distributions of *Chlorella* sp. and *S. obliquus*. *Chlorella* sp. is unicellular and near-spherical with a mean projected area equivalent diameter of 4.57 μm. *S. obliquus* often exists as colonies composed of 4 or 8 cells and is approximately ellipsoid. The projected area equivalent major and minor axes of a single cell average 11.18 μm and 3.70 μm.

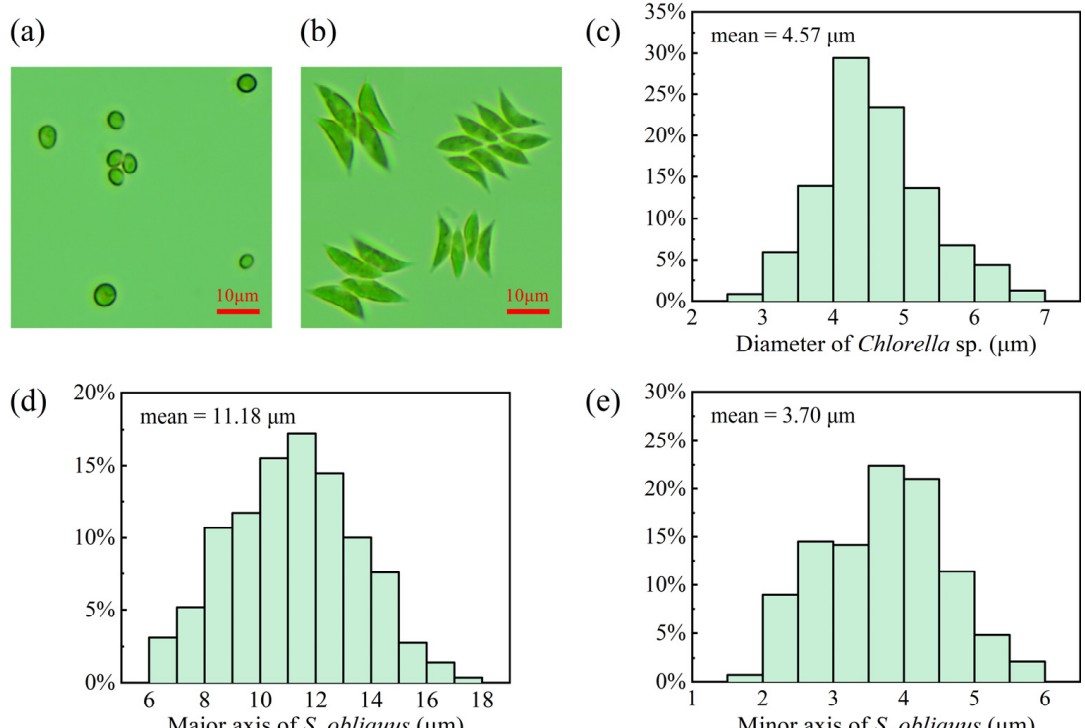

**Figure 4.** Micrographs of (**a**) *Chlorella* sp. and (**b**) *S. obliquus*. Projected area equivalent cell size distribution of (**c**) diameter of *Chlorella* sp., (**d**) major axis and (**e**) minor axis of *S. obliquus*.

Figure 5a is the transmitted light intensity in the PBR during the cultivation of *Chlorella* sp., and that of *S. obliquus* is shown in Supplementary Materials, both of which decrease slowly initially, then fast, and then slowly again. For *Chlorella* sp., the transmitted light

intensity decreased rapidly from day 3 and was hardly detected after day 11. For *S. obliquus*, the transmitted light intensity decreased from day 2 to day 10. Figure 5b is the transmitted light intensity of the PBR with and without aeration every day for *Chlorella* sp., and that for *S. obliquus* is shown in Supplementary Materials. The results show that the transmitted light intensity of the PBR was reduced by bubbles in the initial stage with low microalgae concentrations. In addition, the presence of sparged bubbles did not reduce the transmitted light intensity any longer after day 6 for *Chlorella* sp. and day 4 for *S. obliquus*, meaning that the scattering by bubbles influences the radiation less at higher microalgae concentrations. The corresponding concentration on day 6 for *Chlorella* sp. (0.025 kg·m$^{-3}$) is larger than that on day 4 for *S. obliquus* (0.011 kg·m$^{-3}$), which means that *S. obliquus* attenuates radiation more than *Chlorella* sp. and the effect of bubbles for *S. obliquus* is less than that for *Chlorella* sp. Wheaton and Krishnamoorthy [35] showed that the effect of bubbles was negligible when the microalgae concentrations were over 0.5 kg·m$^{-3}$, and McHardy et al. [36] found that the biomass already counteracted the effects of bubbles at concentrations less than 1 kg·m$^{-3}$. Their results were larger than those in this study, which was mainly caused by the much smaller bubbles or larger aeration ratio (the ratio of gas volume inflowed per min to the culture volume) in their models. However, the research of Luzi et al. [37] indicated that this concentration was also related to the intensity of the light.

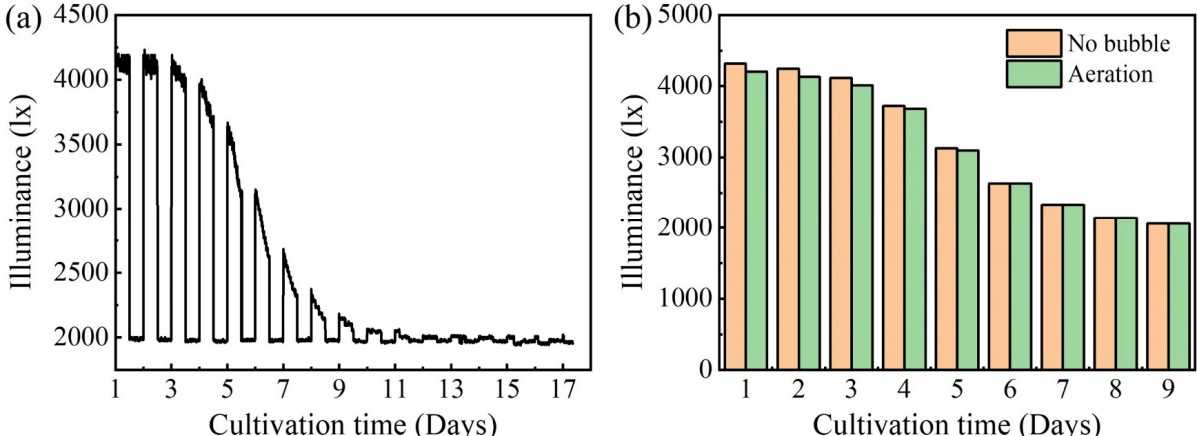

**Figure 5.** (**a**) The transmitted light intensity of the PBR of *Chlorella* sp. with different cultivation times. (**b**) The transmitted light intensity of the PBR with and without aeration every day.

Figure 6 presents the difference between the $CO_2$ concentration of the inlet and outlet gases of the PBR during the cultivation, which equals the amount of $CO_2$ absorbed by the microalgal culture. The results show that the $CO_2$ is almost not absorbed and utilized by microalgae, owing to their slow growth and low microalga concentration. The absorption of $CO_2$ starts on day 2 for *Chlorella* sp. and day 3 for *S. obliquus* and saturates on day 6 for both species of microalgae. The maximal absorption amounts of $CO_2$ are about 0.3% for both species of microalgae, which means that about 30% of $CO_2$ is removed from the 1% $CO_2$ aeration. Chiu et al. [27] reported the efficiency of $CO_2$ reduction in the semicontinuous *Chlorella* sp. cultures was 58%, 27%, 20% and 16% in 2%, 5%, 10% and 15% $CO_2$ aeration, respectively. De Morais and Costa [51] demonstrated that 7–13% of $CO_2$ was fixed by *S. obliquus* at 6% $CO_2$ aeration in a three-serial tubular PBR, and the fixation efficiency decreased to 4–9% at 12% $CO_2$ aeration. The efficiency of $CO_2$ removal or fixation in the PBR is dependent on not only the microalgae species, $CO_2$ concentration, and PBR form [52], but also the $CO_2$ bubble size, which was not mentioned in their research.

Note that the times when little transmitted light is detected and $CO_2$ absorption no longer increases are both earlier than the end of the exponential phase. It suggests that microalgae keep growing rapidly with limited light and $CO_2$. The same behaviors were observed by Tang et al. [53] and Heng and Pilon [22].

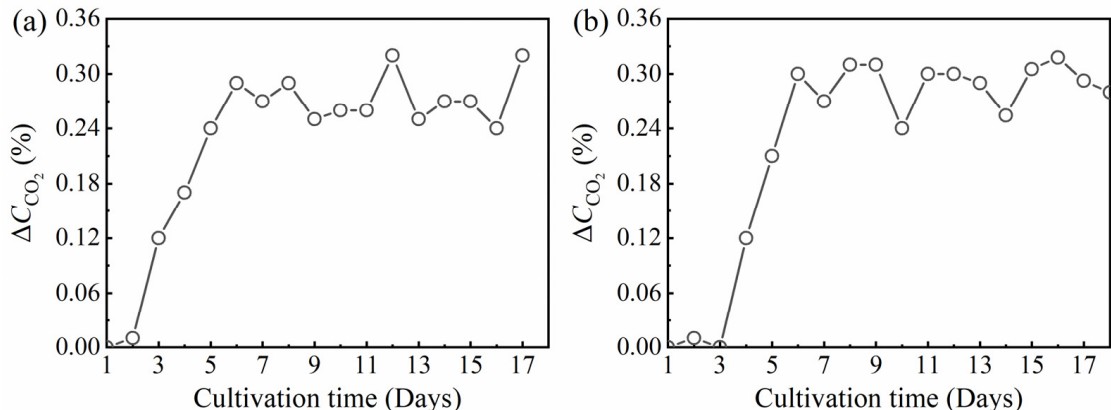

**Figure 6.** The differences in $CO_2$ concentration between the inlet and outlet gas ($\Delta C_{CO2}$) during the cultivation of (**a**) *Chlorella* sp. and (**b**) *S. obliquus*.

Note that the times when little transmitted light is detected and $CO_2$ absorption no longer increases are both earlier than the end of the exponential phase. It suggests that microalgae keep growing rapidly with limited light and $CO_2$. The same behaviors were observed by Tang et al. [53] and Heng and Pilon [22].

### 3.2. Time-Dependent Radiation Characteristics of Microalgae

Figure 7 shows the normal-normal and normal-hemispherical spectral transmittance of the cuvette containing microalgal culture. The results show that the normal-normal and normal-hemispherical transmittances for *Chlorella* sp. decrease rapidly from day 3 to about day 11. By contrast, the transmittance of *S. obliquus* starts to quickly decrease earlier, which is consistent with the decreasing trend of the transmitted light intensity in Figure 5a and Figure S2a. The cell density is extremely low ($<10^5$ cells·ml$^{-1}$ for both species of microalgae) on day 1, so it is believed that the dip in normal-hemispherical transmittance curves over 900 nm is mainly caused by the absorption of the BG11 medium and cuvette glasses, while the microalgal absorption results in the dips around 450 nm and 700 nm as cell density increases.

Figure 8 presents the time-dependent radiation characteristics of the two species of microalgae in the spectral range from 400 to 1100 nm. The absorption cross-sections in Figure 8a,c display peaks at 430 and 680 nm attributed to chlorophyll *a*, at 450 nm and 660 nm attributed to chlorophyll *b*, and at 485 nm attributed to carotenoids [54]. Since drastic fluctuations (even negative values) appeared in cross-sections when microalgal culture was at the lag phase with small cell number density, the absorption and scattering cross-section curves of the first 6 days were omitted from Figure 8. The results show that the absorption cross-sections of both species of microalgae are small in the spectral range over 720 nm. The absorption and scattering cross-sections of *S. obliquus* decreased consistently until day 15, reaching the stationary phase. The maximum values of the absorption cross-section at 430 nm and 680 nm are 2.92 times and 2.58 times the minimum values, respectively, while the absorption and scattering cross-sections of *Chlorella* sp. change relatively slightly during growth. The absorption and scattering cross-sections of *S. obliquus* are larger than those of *Chlorella* sp. because of the larger cell size of *S. obliquus*. The results are consistent with refs. [24,55]. The reason why the absorption and scattering cross-sections decrease with growth time can be explained by the slow synthesis of photosynthetic pigments relative to rapid cell division in the exponential growth phase [24].

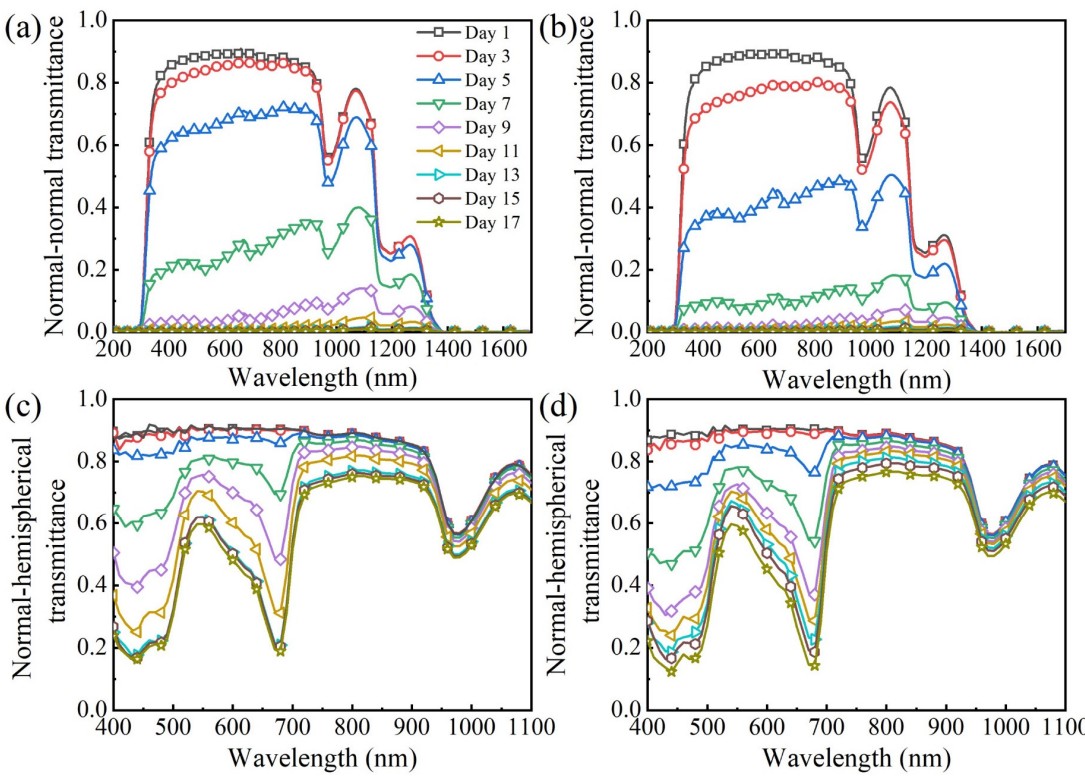

**Figure 7.** (**a**,**b**) Normal-normal and (**c**,**d**) normal-hemispherical spectral transmittance of the cuvette containing *Chlorella* sp. culture and *S. obliquus* culture, respectively.

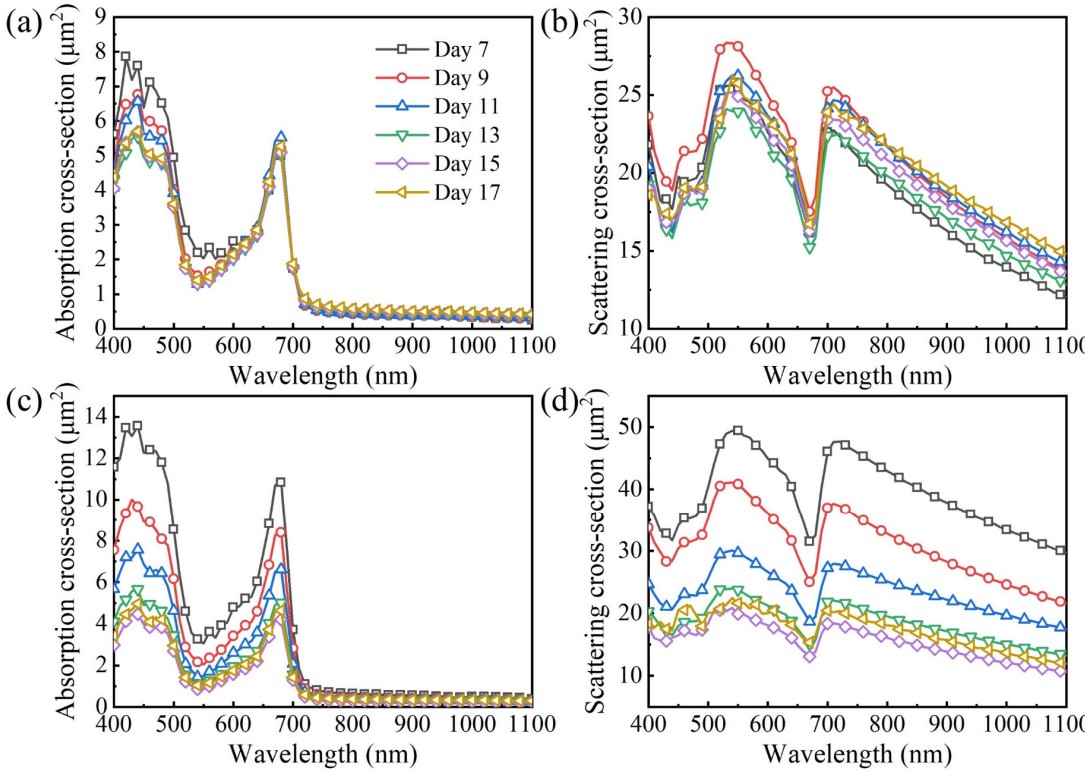

**Figure 8.** Time-dependent radiation characteristics in the spectral range from 400 to 1100 nm. (**a**,**c**) Absorption cross-sections and (**b**,**d**) scattering cross-sections of *Chlorella* sp. and *S. obliquus*, respectively.

In Figure 9, the measured scattering phase function and calculated H-G phase function for *Chlorella* sp. and *S. obliquus* are presented. The scattering phase functions are strongly forward due to the relatively large size parameter and change little with growth time. Therefore, the microalgal scattering phase functions were assumed to be time-invariant. It was found that the H-G phase function provided good approximations of microalgal phase functions. Various species of microalgae scatter light strongly in the forward direction [56]. The asymmetry factor $g_{Xa}$ is generally larger than 0.95 and does not change significantly with wavelength [57].

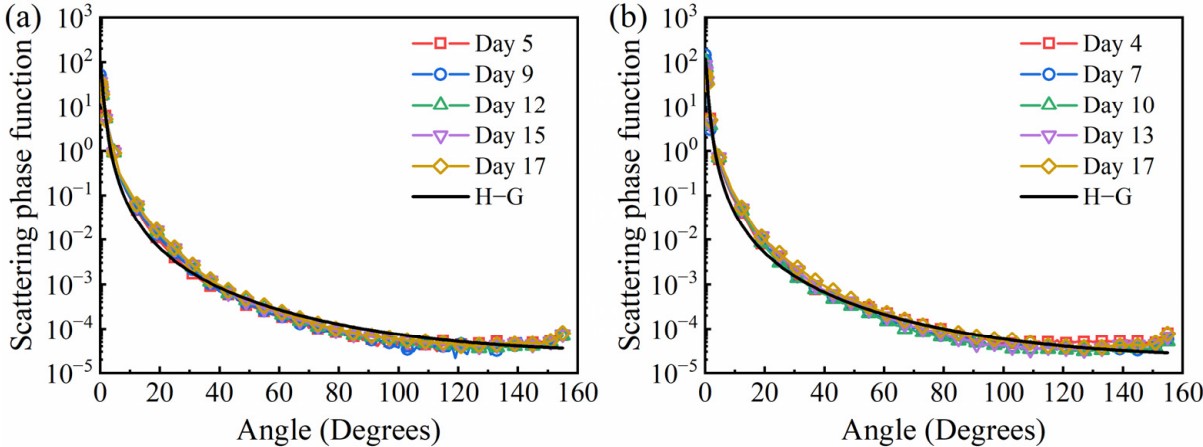

**Figure 9.** Measured and H–G scattering phase functions of (**a**) *Chlorella* sp. ($g_{Xa}$ = 0.98) and (**b**) *S. obliquus* ($g_{Xa}$ = 0.9847).

### 3.3. Time-Dependent Radiation Transfer in the PBR

It is of great significance for microalgal cultivation to simulate the radiation transfer in the PBR during the growth. The time-dependent radiation characteristics of microalgae as well as the optical properties of liquid phase and bubbles are approximated with the box model, as summarized in Supplementary Materials.

Figure 10 shows the one-dimensional radiation distribution in the PBR based on the time-dependent and stationary radiation characteristics, respectively. The results show that there is a relatively small difference for *Chlorella* sp. due to small changes in absorption and scattering cross-sections, while the local irradiance calculated using stationary radiation characteristics was larger than that using time-dependent radiation characteristics since the cross-sections of *S. obliquus* obviously decreased. The results are in agreement with reports by Ma et al. [55] but contrary to Ma et al. [25] as the cross-sections in the former decrease like this study, while those in the latter increase with time. In the early stage of cultivation, the radiation field was slightly affected by microalgae due to the extremely low cell number density. In the late stage of cultivation, the large density of microalgae strongly attenuated the light, and the time-dependent radiation characteristics gradually came close to the value at the stationary phase. As a consequence, the deviation of the local irradiance predicted with the stationary radiation characteristics from that predicted with the time-dependent radiation characteristics first increased and then decreased with cultivation days.

In comparison with Figure 10a,b, it is found that, when calculated with stationary radiation characteristics, the radiation distribution in the PBR of *S. obliquus* was similar to that of *Chlorella* sp. However, when calculated using time-dependent radiation characteristics, the local irradiance in the PBR of *S. obliquus* declines faster than that of *Chlorella* sp., which is consistent with Figure 7, where the spectral transmittance of *S. obliquus* declines faster than that of *Chlorella* sp. The radiation distribution in the PBR varies little after about 13 days for *Chlorella* sp. and 15 days for *S. obliquus*, which is consistent with the time reaching the stationary phase. In addition, the average single scattering albedo is calculated and

given in Supplementary Materials, which contributes to the occurrence of values larger than 1 in Figure 10.

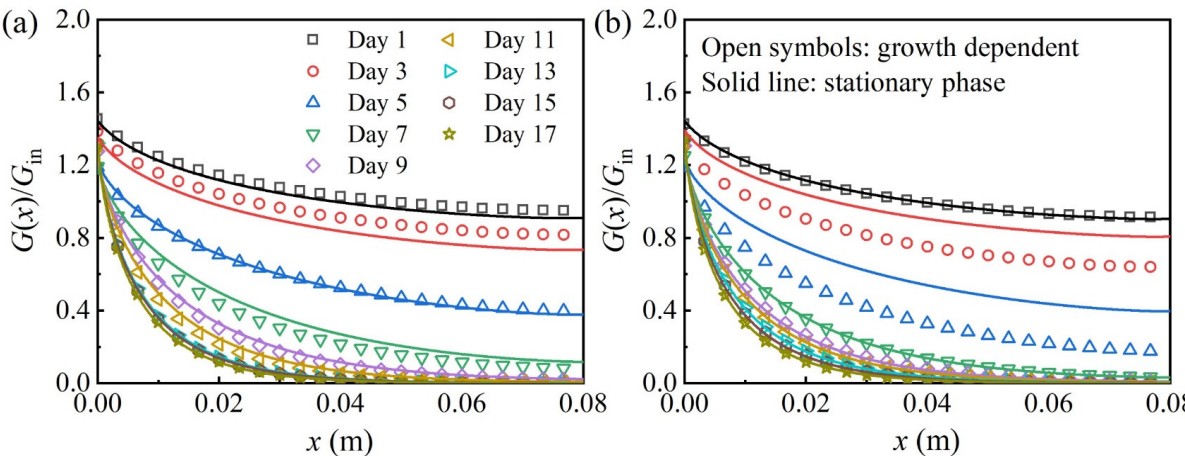

**Figure 10.** Normalized local irradiance $G(x)/G_{in}$ as a function of the distance $x$ from the illuminated culture surface for (**a**) *Chlorella* sp. and (**b**) *S. obliquus*. Open symbols corresponding to time-dependent radiation characteristics, and solid lines corresponding to stationary radiation characteristics (average of last 3 days).

*3.4. Bubble Volume Fraction and Size*

Figure 11 illustrates the effect of bubble volume fraction $f_b$ on the one-dimensional radiation distribution in the PBR with different concentrations of (a,b) *Chlorella* sp. and (c,d) *S. obliquus*. The corresponding average single scattering albedo $\omega_{eff}$ and percent increase in the total irradiance relative to effective incident irradiance at the illuminated culture surface are shown in Table 1. As shown in Figure 11a,c, as $f_b$ increased from 0 to 0.3, the irradiance at the illuminated surface of the culture was enhanced due to bubble scattering, and then it was attenuated more rapidly along with the radiation transfer and might be smaller than that without bubbles at larger culture depths. In Table 1, as $f_b$ increases from 0 to 0.3, the percent increase of the irradiance at the illuminated surface for *Chlorella* sp. in Figure 11a increases from 18.6% to 42.2%, with an increment of 23.6%, and $\omega_{eff}$ increases by 0.185 from 0.745 to 0.930. Those for *S. obliquus* in Figure 11c increase by 18.1% and 0.082, respectively. It is found that the change in the radiation distribution in the PBR of *S. obliquus* is smaller than that of *Chlorella* sp. as $f_b$ increases with low microalgae concentrations, i.e., the effect of bubbles on radiation transfer in the PBR of *S. obliquus* is less than that of *Chlorella* sp. due to the larger absorption and scattering cross-sections of *S. obliquus*. As the $f_b$ increased from 0 to 0.3, the percent increase of surface irradiance and $\omega_{eff}$ for *Chlorella* sp. in Figure 11b increased by 8.2% and 0.017, and those for *S. obliquus* in Figure 11d increased by 9.7% and 0.020, respectively. All increments at different $f_b$ in Figure 11b are smaller than those in Figure 11a for *Chlorella* sp. The same trend occurs in Figure 11c,d for *S. obliquus*, which demonstrates that the bubbles have no obvious effects on the radiation distribution in the PBR at high microalgae concentrations. It is believed that the radiation attenuation is dominated by the absorption of microalgae cells. The same conclusion was drawn by Wheaton and Krishnamoorthy [35].

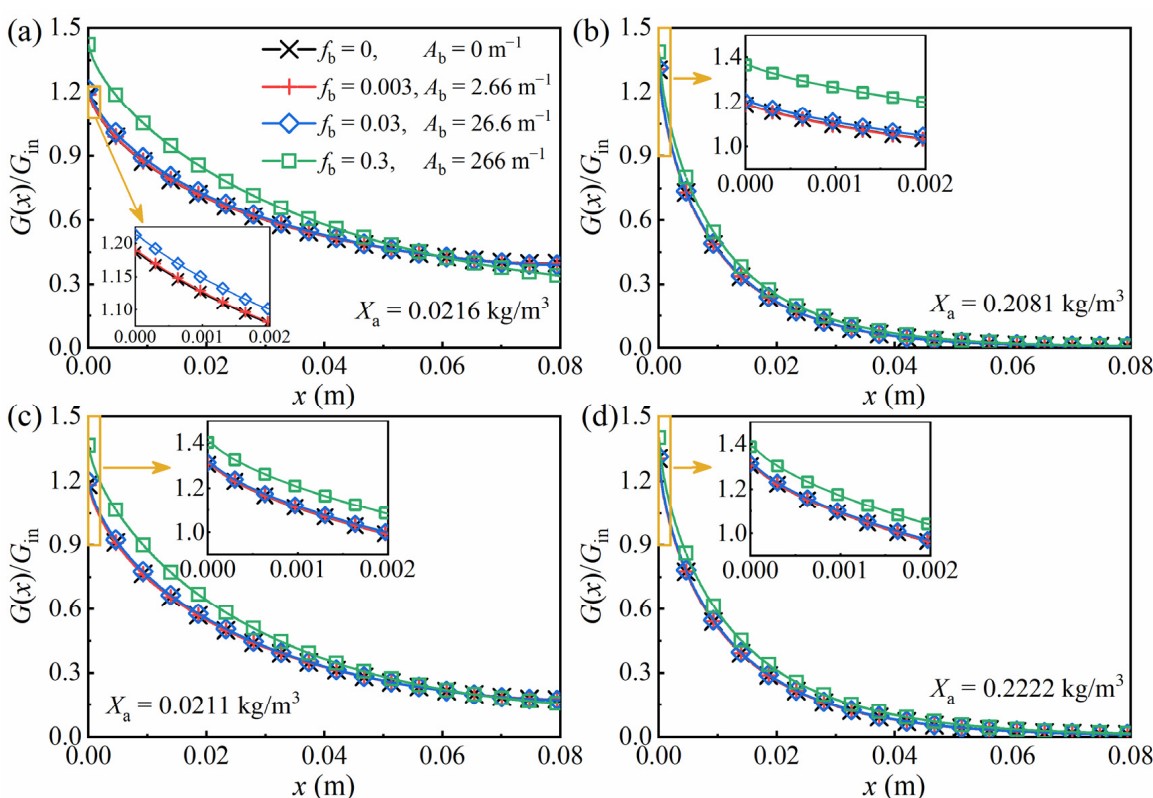

**Figure 11.** Normalized local irradiance $G(x)/G_{in}$ as a function of the distance $x$ from the illuminated culture surface of (**a**,**b**) *Chlorella* sp. and (**c**,**d**) *S. obliquus* for different microalgae concentrations $X_a$ and bubble volume fractions $f_b$ from 0 to 0.3 (bubble radius = 3.5 mm).

**Table 1.** Summary of the average single scattering albedo and percent increase in the total irradiance relative to the effective incident irradiance at the illuminated culture surface of *Chlorella* sp. and *S. obliquus* for bubble volume fractions from 0 to 0.3.

| | $\omega_{eff}$ | | | | $[G(0) - G_{in}]/G_{in}$, % | | | |
|---|---|---|---|---|---|---|---|---|
| | $f_b = 0$ | $f_b = 0.003$ | $f_b = 0.03$ | $f_b = 0.3$ | $f_b = 0$ | $f_b = 0.003$ | $f_b = 0.03$ | $f_b = 0.3$ |
| *Chlorella* sp. ($X_a = 0.0216$ kg/m³) | 0.745 | 0.751 | 0.793 | 0.930 | 18.6 | 18.9 | 21.2 | 42.2 |
| *Chlorella* sp. ($X_a = 0.2081$ kg/m³) | 0.879 | 0.879 | 0.881 | 0.896 | 30.5 | 30.6 | 31.3 | 38.7 |
| *S. obliquus* ($X_a = 0.0211$ kg/m³) | 0.835 | 0.836 | 0.848 | 0.917 | 18.4 | 18.6 | 20.2 | 36.5 |
| *S. obliquus* ($X_a = 0.2222$ kg/m³) | 0.879 | 0.879 | 0.881 | 0.899 | 30.5 | 30.6 | 31.4 | 40.2 |

As shown in Table 1, for $f_b$ = 0, 0.003, or 0.03, $\omega_{eff}$ of two species of microalgal cultures are larger at higher concentrations, while for $f_b$ = 0.3, they are smaller at higher concentrations since the increased microalgae diminish the contribution of bubble scattering to $\omega_{eff}$, which also explains why the percent increase of the surface irradiance of *Chlorella* sp. culture declines from 42.2% to 38.7% in Table 1 ($f_b$ = 0.3). However, $\omega_{eff}$ is not the only factor that affects surface irradiance. As shown in Table 1 ($f_b$ = 0.3), $\omega_{eff}$ for *S. obliquus* in 0.0211 kg/m³ (0.917) is larger than that in 0.2222 kg/m³ (0.899), while the percent increase of the surface irradiance for 0.0211 kg/m³ (36.5%) is less than that for 0.2222 kg/m³ (40.2%) since the back-scattering of microalgae prevails when $\omega_{eff}$ varies slightly. Moreover, $\omega_{eff}$ of the two species of microalgae are not quite the same in low microalgae concentrations and small bubble volume fractions, but there is a tiny difference in the percent increase of the surface irradiance between the two species of microalgae, which is attributed to the small scattering coefficient. The percent increase of local radiation at the illuminated

culture surface in the PBR in this study is generally larger than that in the research of Berberoğlu et al. [32]. Although the microalgae species and bubble size they studied differ from this study, the main reason is that they assume the boundaries to be ideally transparent, while the practical reflectance of the PMMA wall was considered in this study. The trend that the large wall reflectance increases the local irradiance is consistent with the findings of Huang et al. [17].

Figure 12 illustrates the effect of bubble radius $a$ on the one-dimensional radiation distribution in the PBR with different concentrations of (a) and (b) *Chlorella* sp. and (c) and (d) *S. obliquus*. The corresponding average single scattering albedo $\omega_{\text{eff}}$ and percent increase in the total irradiance relative to the effective incident irradiance at the illuminated culture surface are shown in Table 2. It is found that as bubble radius decreases, the trend of the local irradiance in the PBR is similar to that in Figure 11 when $f_b$ increases. The decrease in bubble radius enhances the scattering of a dispersive medium in the PBR, like the increase in $f_b$, so that the local irradiance increases at the illuminated surface of the microalgal culture and is attenuated more rapidly along with the radiation transfer. As radius decreases from 3.5 mm to 3.5 µm, the percent increase in surface irradiance in Figure 12a–d increases by (a) 89.6%, (b) 32.3%, (c) 72.3% and (d) 36.3%, respectively. Figure 12 again indicates that the effect of bubbles on radiation transfer in the PBR of *S. obliquus* is less than that of *Chlorella* sp. and the bubbles have fewer effects on the radiation distribution in the PBR at higher microalgae concentrations. In Table 2 ($a = 3.5$ µm), *S. obliquus* shows the same behavior as *Chlorella* sp. in that the percent increase of surface irradiance is smaller in the higher microalgae concentrations. As the microalgae concentrations increase from 0.0211 kg/m³ to 0.2222 kg/m³ ($a = 35$ µm), $\omega_{\text{eff}}$ decreases a little by 0.013, so the percent increase of surface irradiance still increases. Nevertheless, $\omega_{\text{eff}}$ decreases relatively largely by 0.027 ($a = 3.5$ µm), which affects the surface irradiance more so than the percent increase in surface irradiance decreases.

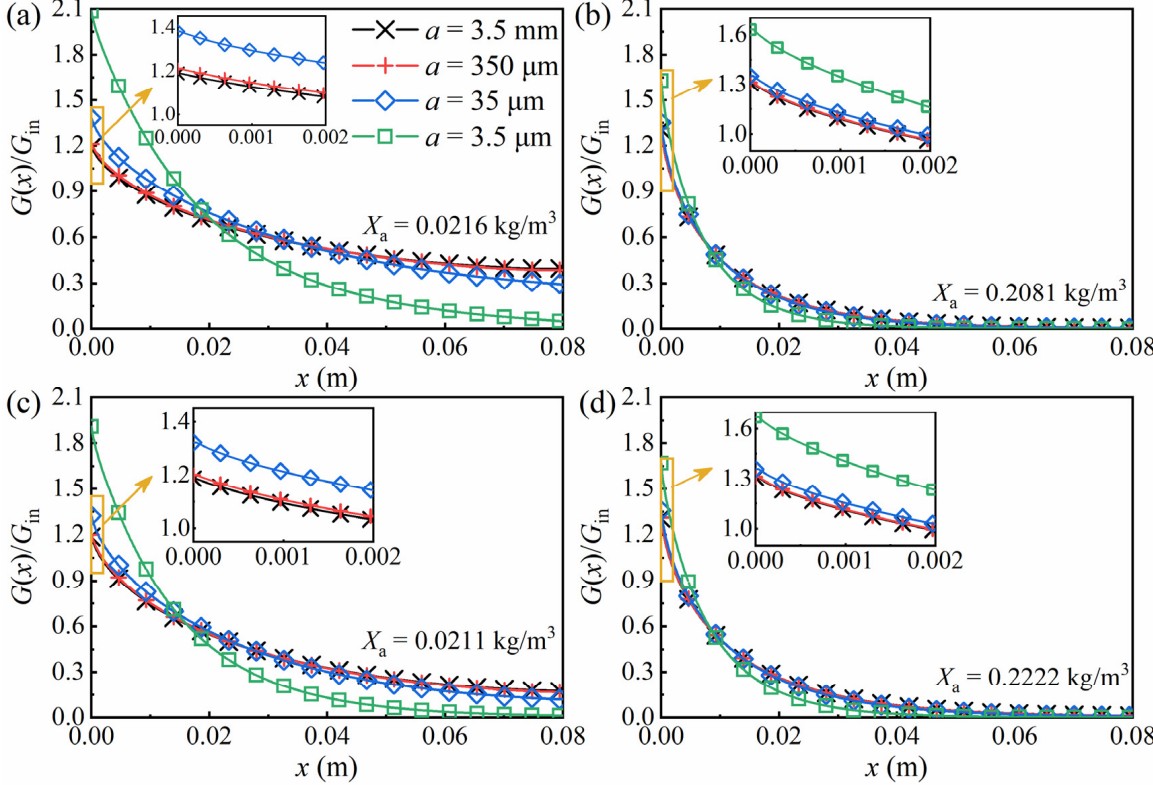

**Figure 12.** Normalized local irradiance $G(\text{x})/G_{\text{in}}$ as a function of the distance $x$ from the illuminated culture surface of (**a**,**b**) *Chlorella* sp. and (**c**,**d**) *S. obliquus* for different microalgae concentrations $X_a$ and for bubble radii $a$ from 3.5 µm to 3.5 mm ($f_b = 0.003$).

**Table 2.** Summary of the average single scattering albedo and percent increase in the total irradiance relative to effective incident irradiance at the illuminated culture surface of *Chlorella* sp. and S. *obliquus* for bubble radii from 3.5 µm to 3.5 mm.

| | $\omega_{eff}$ | | | | $[G(0) - G_{in}]/G_{in}$, % | | | |
|---|---|---|---|---|---|---|---|---|
| | $a$ = 3.5 mm | $a$ = 350 µm | $a$ = 35 µm | $a$ = 3.5 µm | $a$ = 3.5 mm | $a$ = 350 µm | $a$ = 35 µm | $a$ = 3.5 µm |
| *Chlorella* sp. ($X_a$ = 0.0216 kg/m$^3$) | 0.751 | 0.790 | 0.919 | 0.989 | 18.9 | 21.1 | 38.0 | 108.5 |
| *Chlorella* sp. ($X_a$ = 0.2081 kg/m$^3$) | 0.879 | 0.880 | 0.894 | 0.950 | 30.6 | 31.0 | 35.0 | 62.9 |
| S. *obliquus* ($X_a$ = 0.0211 kg/m$^3$) | 0.836 | 0.847 | 0.909 | 0.982 | 18.6 | 20.0 | 32.2 | 90.9 |
| S. *obliquus* ($X_a$ = 0.2222 kg/m$^3$) | 0.879 | 0.881 | 0.896 | 0.955 | 30.6 | 31.1 | 35.8 | 66.9 |

Bubbles with smaller radii and a larger $f_b$ offer larger gas/liquid interfacial area for mass transfer. Berberoğlu et al. [32] found that the percent increase of surface irradiance tends to increase at higher interfacial area concentration $A_b$ (= $3f_b/a$), while they did not investigate the effects of $f_b$ and bubble radius in detail. Comparing Table 2 with Table 1, it is found that in the same $A_b$, the irradiance at the illuminated culture surface with bigger bubbles in Figure 11 is larger than that with smaller bubbles in Figure 12. For example, *Chlorella* sp. cultures in Figure 11a ($f_b$ = 0.3, $a$ = 3.5 mm) and Figure 12a ($f_b$ = 0.003, $a$ = 35 µm) have the same $A_b$ of 266 m$^{-1}$. The percent increase in surface irradiance of the former (42.2%) is larger than that of the latter (38.0%) due to more absorbing volume in the latter, resulting in less light scattered to the illuminated surface. As shown in Figure 12, the smaller the bubbles are, the smaller the local irradiance is at the area near the back surface, which is in agreement with the research of Berberoğlu et al. [32], but opposite to the research of Wheaton and Krishnamoorthy [35]. The reason may be that the former is like this study in terms of PBR's shape and light source installation, while the latter employs cylindrical PBR and sets the light source inside the reactor.

The most common specific growth rate model considering light saturation and inhibition, the Haldane model [58], is used to estimate the effects of bubbles on growth rate and the maximum efficiency of carbon dioxide fixation. The models and identification of parameters, as well as the local specific growth rate, are given in Supplementary Materials, and the average specific growth rates with different $f_b$ and bubble radius are shown in Table 3. The results showed that growth rate decreases as $f_b$ increases (except at $a$ = 3.5 mm) or as radius decreases, since an increase in $f_b$ or a decrease in radius increases the gradient of the light field, resulting in inhibition with excessive irradiance [59] near the illuminated surface and a deficiency of light near the back. While for $a$ = 3.5 mm, the irradiance was mildly and generally enhanced at the entire depth as $f_b$ increased. The maximum growth rate occurred at $f_b$ = 0.3 and $a$ = 3.5 mm in all cases. While $f_b$ = 0.003 and $a$ = 3.5 mm is recommended as the optimal condition considering the trade-off between increased growth rate and cost. The opposite effect of bubble radius on growth rate between this study and the literature [30] is not unacceptable since small bubbles improve the growth of microalgae through the promotion of mass transfer [11]. The results in this study indicate that the volume fraction and bubble radius should be simultaneously considered in radiation transfer rather than summarized as the interfacial area concentration.

**Table 3.** Average specific growth rate (d$^{-1}$) different bubble volume fraction and radius.

| | $f_b = 0.003$ | | | | $f_b = 0.03$ | | | | $f_b = 0.3$ | | | |
|---|---|---|---|---|---|---|---|---|---|---|---|---|
| | $a = 3.5$ mm | $a = 350$ μm | $a = 35$ μm | $a = 3.5$ μm | $a = 3.5$ mm | $a = 350$ μm | $a = 35$ μm | $a = 3.5$ μm | $a = 3.5$ mm | $a = 350$ μm | $a = 35$ μm | $a = 3.5$ μm |
| *Chlorella* sp. ($X_a = 0.0216$ kg/m$^3$) | 1.6492 | 1.6468 | 1.6107 | 1.2296 | 1.6507 | 1.6139 | 1.2284 | 0.5322 | 1.6614 | 1.2714 | 0.5480 | 0.2325 |
| *Chlorella* sp. ($X_a = 0.2081$ kg/m$^3$) | 0.6412 | 0.6386 | 0.6144 | 0.4729 | 0.6457 | 0.6208 | 0.4760 | 0.2307 | 0.7010 | 0.5302 | 0.2522 | 0.1207 |
| *S. obliquus* ($X_a = 0.0211$ kg/m$^3$) | 0.5978 | 0.5978 | 0.5872 | 0.4483 | 0.6013 | 0.5905 | 0.4490 | 0.2056 | 0.6349 | 0.4789 | 0.2163 | 0.0905 |
| *S. obliquus* ($X_a = 0.2222$ kg/m$^3$) | 0.1841 | 0.1837 | 0.1793 | 0.1504 | 0.1858 | 0.1813 | 0.1516 | 0.0850 | 0.2057 | 0.1696 | 0.0929 | 0.0441 |

## 4. Conclusions

The radiation transfer in PBRs is greatly affected by the microalgal time-dependent radiation characteristics of microalgae and bubbles. A thorough analysis of radiation transfer during the microalga growth in PBRs is essential for improving the cultivation. This study reports the temporal evolution of the scattering and absorbing cross-sections for *Chlorella* sp. and *S. obliquus*, which are grown in the airlift flat plate PBR. A one-dimensional model for radiation transfer in the PBR was proposed, considering the time-dependent radiation characteristics and bubbles. The scattering and absorption cross-sections of *S. obliquus* are found to decrease consistently over cultivation time, while those of *Chlorella* sp. change slightly. For *S. obliquus*, the local irradiance calculated using the radiation characteristics of the stationary phase is larger than that using time-dependent radiation characteristics. For *Chlorella* sp., the deviation is relatively small. With the increase of bubble volume fraction or the decrease of bubble radius, the local irradiance increases at the illuminated surface of the microalgal culture and is attenuated more rapidly along with the radiation transfer. The effect of bubbles on radiation transfer in the PBR of *S. obliquus* is less than that of *Chlorella* sp. The bubbles have no obvious effects on the radiation distribution in the PBR at high microalgae concentrations. It is believed that the radiation attenuation is dominated by the absorption of microalgae cells. The irradiance at the illuminated culture surface is related to various factors, including average single scattering albedo and boundary reflection. In the same interfacial area concentration, the irradiance at the illuminated culture surface with bigger bubbles is higher than that with smaller bubbles. The average specific growth rate decreases as bubble volume fraction increases or bubble radius decreases, resulting from the increased gradient of the light field. The volume fraction of 0.003 and a radius of 3.5 mm are the optimal operating conditions in this study for microalgae growth and carbon dioxide fixation. The presented analysis will facilitate the design and optimization of the optical and aeration configurations of PBRs to achieve a high growth rate and promote carbon dioxide fixation.

**Supplementary Materials:** The following supporting information can be downloaded at: https://www.mdpi.com/article/10.3390/photonics9110864/s1, Figure S1: The spectral irradiance of LED lights [60–63]; Figure S2: (**a**) The transmitted light intensity of the PBR of *S. obliquus* with respect to cultivation days. (**b**) The transmitted light intensity of the PBR with and without aeration every day; Figure S3: (**a**) The spectral mass scattering and absorption cross-sections of *Chlorella* sp. on day 13. (**b**) The spectral absorption coefficient of water; Figure S4: Temporal evolutions of average single scattering albedo of (**a**) *Chlorella* sp. and (**b**) *S. obliquus*; Figure S5: Specific growth rates of experimental data and kinetic model fitting with different irradiance [58,64]; Figure S6: Local specific growth rates for different bubble volume fractions and radii of (**a,c,e**) *Chlorella* sp. and (**b,d,f**) *S. obliquus*.; Table S1: BG11 (Blue-Green Medium) [41]; Table S2: A5 (Trace mental solution); Table S3: Summary of the optical properties of *Chlorella* sp. on day 13 and boundary conditions in the box model. [32,48,49,65].

**Author Contributions:** Conceptualization, X.L., J.-Y.Y. and L.L. (Linhua Liu); methodology, T.F. and L.L. (Li Lin); software, L.L. (Li Lin); validation, J.Z. and L.L. (Linhua Liu); resources, X.L. and J.Z.; data curation, T.F. and L.L. (Li Lin); writing—original draft preparation, T.F.; writing—review and editing, X.L., J.-Y.Y. and L.L. (Linhua Liu); supervision, X.L., J.-Y.Y., J.Z. and L.L. (Linhua Liu); project administration, X.L., J.-Y.Y. and L.L. (Linhua Liu). All authors have read and agreed to the published version of the manuscript.

**Funding:** This research was funded by the National Natural Science Foundation of China, grant number 52106080 and 52076123.

**Data Availability Statement:** Data underlying the results presented in this paper are not publicly available at this time but may be obtained from the authors upon reasonable request.

**Acknowledgments:** We are grateful to Haiyan Pei (Shandong Provincial Engineering Center on Environmental Science and Technology) for providing the cultures of *Chlorella* sp. and *S. obliquus*.

**Conflicts of Interest:** The authors declare no conflict of interest.

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
