# Peer review of "Modeling Effect of Bubbles on Time-Dependent Radiation Transfer of Microalgae in a Photobioreactor for Carbon Dioxide Fixation"

_photonics, doi:10.3390/photonics9110864_

Round 1
Reviewer 1 Report
The authors carried out one-dimensional numerical simulations of the steady-state radiative transfer equation considering the presence of the dispersed volume fraction and confronted the numerical results with the experimental ones. I am in favor of the publication of the present manuscript if the following issues are addressed.
1) There are a couple of interesting manuscripts that deal with integrated numerical simulations of fluid flow hydrodynamics, light distribution, and growth kinetics in a photobioreactor. Those are: “Numerical analysis of the effects of air on light distribution in a bubble column photobioreactor”, Algal Research, Volume 31, April 2018, Pages 311-325, and “Comparison between different strategies for the realization of flashing-light effects – Pneumatic mixing and flashing illumination”, Algal Research, Volume 38, March 2019, 101404. They should be mentioned in the Introduction after Line 90 together with the work of Wheaton and Krishnamoorthy [35] to have a wider overview of the work already done in the simulation field.
2) Line 238-240. The results should be also compared with the findings of the two manuscripts mentioned in point 1.
3) Page 11-19. Figure 11 is displayed multiple times and in a chaotic format. Please, adjust the format and show Figure 11 only once.
4) In Figure 11 the variation of the normalized local irradiance as a function of the distance from the illuminated culture surface for the case where the volume fraction is not considered, i.e., fb=0 is barely visible. The authors should find a way to make Figure 11 clear.
5) Line 382-384. The authors wrote “all the increments in Figure 11b) and Figure 11d) are smaller than those exhibited in Figure 11a) and Figure 11c)..”. The authors should point out that the results of Figures 11a) and c) are obtained for a different specie than the results of Figures 11b) and d).
6) Figure 12, line 460. In Figure 12 a) and c) the insets are missing. Besides, the authors should find a way to visualize the curves corresponding to the case a=3.5 mm (the black ones). Moreover, in Figure 12 a) Ab is not needed since it is already written in line 451.
Author Response
The authors carried out one-dimensional numerical simulations of the steady-state radiative transfer equation considering the presence of the dispersed volume fraction and confronted the numerical results with the experimental ones. I am in favor of the publication of the present manuscript if the following issues are addressed.
Response: Great thanks for your treasurable comments. We will revise this manuscript to meet the high standards of Photonics.
1) There are a couple of interesting manuscripts that deal with integrated numerical simulations of fluid flow hydrodynamics, light distribution, and growth kinetics in a photobioreactor. Those are: “Numerical analysis of the effects of air on light distribution in a bubble column photobioreactor”, Algal Research, Volume 31, April 2018, Pages 311-325, and “Comparison between different strategies for the realization of flashing-light effects – Pneumatic mixing and flashing illumination”, Algal Research, Volume 38, March 2019, 101404. They should be mentioned in the Introduction after Line 90 together with the work of Wheaton and Krishnamoorthy [35] to have a wider overview of the work already done in the simulation field.
Response: Great thanks for the good comment. We have read these papers and cited them in the Introduction as “McHardy et al. [36] numerically investigated the impact of gas bubbles on light distribution in a bubble column PBR under different gas flow rate and microalgae concentrations. Luzi et al. [37] evaluated the enhancement of culture growth by pulsed illumination and pneumatic mixing in a bubble column PBR through numerical simulations.”
2) Line 238-240. The results should be also compared with the findings of the two manuscripts mentioned in point 1.
Response: Great thanks for the good comment. We have compared our results with the findings of the references and added to the corresponding place of the main text as “Wheaton and Krishnamoorthy [35] showed that the effect of bubbles was negligible when the microalgae concentrations were over 0.5 kgm-3, and McHardy et al. [36] found the biomass counteracts effects of bubbles already at concentrations less than 1 kgm-3.Their results were larger than that in this study, which is mainly caused by the much smaller bubbles or larger aeration ratio (the ratio of gas volume inflowed per min to the culture volume) in their models. However, the research of Luzi et al. [37] indicated that this concentration is also related to the intensity of the light.”
3) Page 11-19. Figure 11 is displayed multiple times and in a chaotic format. Please, adjust the format and show Figure 11 only once.
Response: Great thanks. We have revised the Figures 11 and 12 for clarity. We changed the symbols of lines and merged the four subfigures.
4) In Figure 11 the variation of the normalized local irradiance as a function of the distance from the illuminated culture surface for the case where the volume fraction is not considered, i.e., fb=0 is barely visible. The authors should find a way to make Figure 11 clear.
Response: Great thanks for the comment. We have revised the Figures 11 for clarity. We changed the symbols of lines and removed repetitive legends.
5) Line 382-384. The authors wrote “all the increments in Figure 11b) and Figure 11d) are smaller than those exhibited in Figure 11a) and Figure 11c)..”. The authors should point out that the results of Figures 11a) and c) are obtained for a different specie than the results of Figures 11b) and d).
Response: Great thanks. We are sorry that we didn't point out the species of microalgae, which may puzzle the reader. The sentence has been revised for clarity as “All increments at different fb in Figure 11(b) are smaller than those in Figures 11(a) for Chlorella sp. The same trend occurs in Figure 11(d) and (c) for S. obliquus, which demonstrates that the bubbles have no obvious effects on the radiation distribution in the PBR at high microalgae concentrations.”
6) Figure 12, line 460. In Figure 12 a) and c) the insets are missing. Besides, the authors should find a way to visualize the curves corresponding to the case a=3.5 mm (the black ones). Moreover, in Figure 12 a) Ab is not needed since it is already written in line 451.
Response: Great thanks for the comment. We have revised the Figure 12 and we have added the insets and removed the Ab.

Reviewer 2 Report
Review of the manuscript "Effect of bubbles on time-dependent radiation transfer of microalgae in a photobioreactor for carbon dioxide fixation", by Tianhao Fei, Li Lin, Xingcan Li, Jia-Yue Yang, Junming Zhao, and Linhua Liu
In the proposed manuscript, the authors studied the effects of bubble size and volume fraction on radiation transfer in the photobioreactor. To ensure the accuracy of the results, a model considering bubble scattering was built and the time-dependent radiation characteristics of microalgae were obtained through well-setting experiments. From my point of view, the article is a valuable addition to the field of radiative transfer in the photobioreactors. The article is well written and the methods applied are appropriate. Clear figures are given to make the paper understood easily. I think this article is helpful to the readers of Photonics and suggest some minor modifications before publication.
The following aspects need to be more detailed to increase the scientific quality of the manuscript:
Section 2.1, line 119: The word "measured" is redundant and can be deleted.
Section 2.2, How much culture was taken away for measurement during the whole experiment? Will it affect the number density of microalgae in the photobioreactor?
Section 2.2, the absorption coefficients formula Eq. (2) should be explained and verified, i.e., authors should carefully use this forlmula? The formula of absorption coefficient should not the simple analogy with that of the extinction coefficients, due to the multiple scattering or oblique light with the container surface existed in the measurement process.
Section 3.1, In Figure 5(b), why the illuminance is only presented till day 9?
Section 3.2, Why does the spectrum ranged from 400 to 1100 nm? Microalgae only use photons between 400 and 750 nm for photosynthesis.
Section 3.2, line 281 to 283, the authors said, “drastic fluctuations (even negative values) appeared in cross-sections”, Does the author analyze the reasons for this result? Maybe try to use the data processing method proposed by Pilon's research group? How much does the processing method (Eq.(1) to (4) ) improve the accuracy of the coefficient/cross-section data compare with Pilon’s method?
How the optical properties of bubbles were obtained should be stated? Was the Mie theory used or other method? Moreover, the total scattering phase function should be the bubbles plus the microalgae cells?
In addition, the one-dimensional model on radiation transfer in photobioreactor should be clearly stated? What is the numerical method used for solving the RTE?
My conclusion - The proposed paper requires minor revisions.
Author Response
In the proposed manuscript, the authors studied the effects of bubble size and volume fraction on radiation transfer in the photobioreactor. To ensure the accuracy of the results, a model considering bubble scattering was built and the time-dependent radiation characteristics of microalgae were obtained through well-setting experiments. From my point of view, the article is a valuable addition to the field of radiative transfer in the photobioreactors. The article is well written and the methods applied are appropriate. Clear figures are given to make the paper understood easily. I think this article is helpful to the readers of Photonics and suggest some minor modifications before publication.
Response: Great thanks. We will revise this manuscript to meet the high standards of Photonics.
Section 2.1, line 119: The word "measured" is redundant and can be deleted.
Response: Thanks. We have removed the redundant word.
Section 2.2, How much culture was taken away for measurement during the whole experiment? Will it affect the number density of microalgae in the photobioreactor?
Response: We thank the reviewer for the good comment. The sampling volume was about 500 ml per day, which is rather small and had little effect on the culture with a much larger volume of 24.6 L. Meanwhile, same volume of fresh and sterilized BG11 medium was added back to the culture after sampling. Therefore, the effects of daily sampling on growth of microalgae including number density can be neglected.
Section 2.2, the absorption coefficients formula Eq. (2) should be explained and verified, i.e., authors should carefully use this formula? The formula of absorption coefficient should not the simple analogy with that of the extinction coefficients, due to the multiple scattering or oblique light with the container surface existed in the measurement process.
Response: We thank the reviewer for the good comment. The transmission method for determinations of extinction and absorption coefficients is based on the Beer–Lambert’s law, which is only different in acceptance angles of the detector for the two coefficients. Indeed, ideal normal transmissions are impossible to detect due to the certain size of detector window, while the hemispherical transmissions can be relatively ideally detected using the integrating sphere (system error is still exists because small areas of reflection surface are replaced by sample and detector windows). In our measurements, the detector diameter is 3 mm and the distance between the detector and cuvette is about 300 mm, so the acceptance angle is about 0.58°, which is small enough. And the areas of sample and detector windows is less than 1% of the total inner surface of the integrating sphere. The scattering is assumed to be single due to the quite small volume fraction of microalgae (generally < 0.001) in our experiments. Although there is oblique light with the container surface, the three-layer structure can be equivalently divided into three single-layer models, assuming that the incident light of every layer is collimated since glass layer is non-scattering and non-absorption medium. The scattering by microalgae is strongly forward, which also decreases the effect of oblique light. Pilon's group [1] also used the integrating sphere to determine the absorption coefficient, which is based on the similar fundamental principles as ours. The difference is that they used a reference medium to correct the reflection by cuvette, while we calculated the reflection of glass layer with optical properties. In addition, the similar approximation, assuming non-straight light travelling in the dispersive medium to be in two directions (forward and reverse), is extensively applied in two-flux theory [2,3] with similar differential equation to Beer–Lambert’s law.
References:
1 Heng, R.-L.; Pilon, L. Time-dependent radiation characteristics of Nannochloropsis oculata during batch culture. J. Quant. Spectros. Radiat. Transfer 2014, 144, 154-163.
2 Chen, S.; Yuan, L.; Weng, X.; Deng, L. Modeling emissivity of low-emissivity coating containing horizontally oriented metallic flake particles. Infrared Physics & Technology 2014, 67, 377-381.
3 Vargas, W.E. Inversion methods from Kiabelka-Munk analysis. Journal of Optics a-Pure and Applied Optics 2002, 4, 452-456.
Section 3.1, In Figure 5(b), why the illuminance is only presented till day 9?
Response: Great thanks. As shown in Figure 5(b), The transmitted light intensity of the PBR with aeration is the same as that without bubbles after day 6 for Chlorella sp. and day 4 for S. obliquus, and the illuminance value (including days after 9th) can be obtained in Figure 5(a). To avoid repetition, the illuminance is presented till day 9 in Figure 5(b).
Section 3.2, Why does the spectrum range from 400 to 1100 nm? Microalgae only use photons between 400 and 750 nm for photosynthesis.
Response: We thank the reviewer for the good comment. It is true that microalgae only use photons between 400 and 750 nm for photosynthesis. There is no problem in selecting spectrum larger. The measuring instrument employed is available at 400-1100 nm, and as shown in Figure 8, the absorption cross-sections of microalgae over 720 nm is very small. Therefore, the presence of 720-1100 nm band has little influence on the analysis and conclusion of this study, and gives an additional information of spectral radiation characteristics.
Section 3.2, line 281 to 283, the authors said, “drastic fluctuations (even negative values) appeared in cross-sections”, Does the author analyze the reasons for this result? Maybe try to use the data processing method proposed by Pilon's research group? How much does the processing method (Eq.(1) to (4) ) improve the accuracy of the coefficient/cross-section data compare with Pilon’s method?
Response: We thank the reviewer for the good comment. The weaker signal and lower instrument precision result in bigger fluctuations of data. There are a few little fluctuations in the hemispherical transmittance below 550 nm, which is obtained by the integrating sphere (model RTC-060-IG; Labsphere, Inc., USA). In the calculation of absorption cross-sections of microalgae, the fluctuations will be magnified when dividing the absorption coefficients by rather small cell number density at lag phase of growth. When the number density increases day by day, which is hundreds or thousands of times the beginning, making the fluctuations decrease. As mentioned above, our processing method (Eq.(1) to (4) ) is based on the similar fundamental principles as Pilon's. So, the applicable conditions and the accuracy of the two methods are similar. However, they used a reference medium to correct the reflection of cuvette, which only consider the first-order transmission of the three layers. While we calculated the transmission of the three-layer model considering the higher-order transmission. According to the work of Li et al. [4], the transmission method used in our works is more accurate than Pilon's to some extent.
References:
4 Li, X.C.; Zhao, J.M.; Wang, C.C.; Liu, L.H. Improved transmission method for measuring the optical extinction coefficient of micro/nano particle suspensions. Appl. Opt. 2016, 55, 8171-8179.
How the optical properties of bubbles were obtained should be stated? Was the Mie theory used or other method? Moreover, the total scattering phase function should be the bubbles plus the microalgae cells?
Response: We thank the reviewer for the good comment. The scattering efficiency factor Qsca,b and the scattering phase function of the bubbles Φb are predicted by Mie theory applied to a sphere of radius a (3.5 μm, 35 μm, 350 μm, 3.5 mm) and refractive index 1 embedded in water with nL=1.33. The results indicate that Qsca,b is equal to 1.0 (corrected for the diffraction paradox) and it generally varies less than 0.1% (a = 3.5 mm), 0.2% (a = 350 μm), 0.5% (a = 35 μm), 5% (a = 3.5 μm) in the considering spectrum. Similarly, it was found that Φb is strongly forward and does not vary appreciably for the size parameters considered. The asymmetry factors gb are 0.84600 (a = 3.5 mm, deviation < 0.04%), 0.84613 (a = 350 μm, deviation < 0.2%), 0.85087 (a = 35 μm, deviation < 0.5%), and 0.85597 (a = 3.5 μm, deviation < 1.5%) in the considering spectrum. In order to simplify the calculations, the phase function of the bubble is approximated as H-G phase function with gb obtained by Mie theory. Finally, the total scattering phase function is the sum of bubbles and microalgae cells:
|
|
This part has been revised and added in Section 2 of Supplementary materials.
In addition, the one-dimensional model on radiation transfer in photobioreactor should be clearly stated? What is the numerical method used for solving the RTE?
Response: We thank the reviewer for the good comment. The finite volume method was employed to discretely solve the RTE, which has been widely applied to various radiative heat transfer problems since the work of Raithby and Chui [5]. The spatial discretization, boundary condition and mathematical process can be seen in many books [6,7] and journals[8-10], and will not be repeated in this paper. We have added relevant quotations in Section 2.3 of main text and gave a brief description in the Section 6 of Supplementary materials.
References:
5 Raithby, G.D.; Chui, E.H. A finite-volume method for predicting a radiant-heat transfer in enclosures with participating media. Journal of Heat Transfer-Transactions of the Asme 1990, 112, 415-423.
6 Čanić, S.; Delle Monache, M.L.; Piccoli, B.; Qiu, J.M.; Tambača, J. Chapter 16 - Numerical Methods for Hyperbolic Nets and Networks. In Handbook of Numerical Analysis; Abgrall, R.; Shu, C.-W., Eds.; Elsevier: 2017; Volume 18, pp. 435-463.
7 Mazumder, S. Chapter 7 - Unstructured Finite Volume Method. In Numerical Methods for Partial Differential Equations; Mazumder, S., Ed.; Academic Press: 2016; pp. 339-388.
8 Kim, S.H.; Huh, K.Y. A new angular discretization scheme of the finite volume method for 3-D radiative heat transfer in absorbing, emitting and anisotropically scattering media. Int. J. Heat Mass Transfer 2000, 43, 1233-1242.
9 Hao, J.B.; Luan, L.M.; Tan, H.P. Effect of anisotropic scattering on radiative heat transfer in two-dimensional rectangular media. J. Quant. Spectros. Radiat. Transfer 2003, 78, 151-161.
10 Chai, J.C. One-dimensional transient radiation heat transfer modeling using a finite-volume method. Numerical Heat Transfer, Part B: Fundamentals 2003, 44, 187-208.

Round 2
Reviewer 1 Report
Dear Sir/Madam,
after carefully checking the revised manuscript entitled "Modeling effect of bubbles on time-dependent radiation transfer of microalgae in a photobioreactor for carbon dioxide fixation" I recognised that the quality of the manuscript significantly improved. I have only the following minor concern:
1) Page 11, Lines 368-370. I suggest to remove the values 0.0216 kg/m3 and 0.0211 kg/m3.
Best regards
Author Response
After carefully checking the revised manuscript entitled "Modeling effect of bubbles on time-dependent radiation transfer of microalgae in a photobioreactor for carbon dioxide fixation" I recognised that the quality of the manuscript significantly improved. I have only the following minor concern:
Response: Great thanks for your treasurable comments. We will revise this manuscript to meet the high standards of Photonics.
1) Page 11, Lines 368-370. I suggest to remove the values 0.0216 kg/m3 and 0.0211 kg/m3.
Response: Great thanks. We have removed the “(0.0216 kg/m3)”and “(0.0211 kg/m3)” in the corresponding text.